# Modified U Slot Patch Antenna with Large Frequency Ratio for Vehicle-to-Vehicle Communication

**DOI:** 10.3390/s23136108

**Published:** 2023-07-03

**Authors:** Sandhiya Reddy Govindarajulu, Md Nurul Anwar Tarek, Marisol Roman Guerra, Asif Hassan, Elias Alwan

**Affiliations:** Department of Electrical and Computer Engineering, Florida International University, 10555 West Flagler St. EC 3900, Miami, FL 33199, USA; sredd015@fiu.edu (S.R.G.); mroma078@fiu.edu (M.R.G.); ahass046@fiu.edu (A.H.)

**Keywords:** coupled feed, DSRC, dual-band patch, millimeter-wave, vehicle-to-vehicle communication

## Abstract

This paper presents a single-fed, single-layer, dual-band antenna with a large frequency ratio of 4.74:1 for vehicle-to-vehicle communication. The antenna consists of a 28 GHz inset-fed rectangular patch embedded into a 5.9 GHz patch antenna for dual-band operation. The designed dual-band antenna operates from 5.81 to 5.99 GHz (Dedicated Short Range Communications, DSRC) and 27.9 to 30.1 GHz (5G millimeter-wave (mm-wave) band). Furthermore, the upper band patch was modified by inserting slots near the inset feed line to achieve an instantaneous bandwidth of 4.5 GHz. The antenna was fabricated and measured. The manufactured prototype operates simultaneously from 5.8 to 6.05 GHz and from 26.8 to 31.3 GHz. Notably, the designed dual-band antenna offers a high peak gain of 7.7 dBi in the DSRC band and 6.38 dBi in the 5G mm-wave band.

## 1. Introduction

Vehicle-to-Vehicle communication (V2V) is the future of the Intelligent Transportation System [1]. The Federal Communication Commission (FCC) originally allocated the Dedicated Short Range Communications (DSRC) band (i.e., 5.85 to 5.925 GHz) for V2V. However, this band is not sufficient for high-data-rate transmission. Conversely, the millimeter-wave (mm-wave) spectrum offers a larger bandwidth for high-speed communication [2]. Therefore, a system with a large frequency ratio operating at microwave and mm-wave frequencies is paramount to supporting the next-generation transportation system.

Multiband systems use separate antennas, each operating at a specific band, or a single antenna structure resonating at various bands. Examples of dual-band single-fed planar antennas include patch antennas [3,4], co-planar waveguide (CPW)-fed slot antennas [5], and planar inverted-F antennas (PIFA) [6]. While these dual-band antennas are compact, their frequency ratio (i.e., the ratio of the center frequency of the upper band to the lower band) is small, with most being less than 3.

A simple way to realize dual-band antennas with a large frequency ratio is to integrate two different frequency antennas on the same aperture [7,8,9,10]. Multiband antennas on the same aperture reduce the cost and weight of the system. A multilayer dual-band antenna with a magneto-electric dipole and parallel-plate resonator on different layers operating at 5.9 GHz and 28 GHz was developed in [11]. However, the design requires expensive multilayer fabrication and suffers from a high profile (i.e., $h=0.12\lambda_{l}$).

Indeed, designing a single-layer single-port dual-band antenna with a large frequency ratio is daunting due to the large difference in antenna dimensions. In ref. [12], a multiband antenna with monopole and patch antenna separated by a low-pass filter for 2.4 GHz, 5.5 GHz, and 28 GHz applications was designed. However, the gain at the lower bands remains low, with 1.95 dBi at 2.4 GHz and 3.76 dBi at 5.5 GHz.

In ref. [13], a microstrip grid array antenna with parasitic patches and a differential feeding network was designed to achieve dual-band operation. However, the parasitic patches increase the overall dimension of the antenna. In ref. [14], a single-fed dual-band antenna with microwave patches operating at 4.85 GHz and a stub-loaded microstrip line operating at 26 GHz was implemented. This antenna has a high-profile design and requires an additional feeding structure design. In ref. [15], a structure sharing a single-fed substrate integrated waveguide (SIW) slot antenna with a monopole was reported. However, this antenna provides a low gain of 2.6 dBi at 2.4 GHz, and the structure is complicated.

In [16], a stacked patch structure is used to achieve resonance in the 5.8 GHz band, while two arm patches achieve resonance in the 2.4 GHz band. However, this designed antenna achieves a frequency band ratio of approximately 2.4:1 and only a 13% bandwidth in the 5.5 GHz band. In ref. [17], the microwave parallel-plate waveguide resonator and mm-wave Fabry-Perot resonator were cleverly combined to provide a dual-frequency antenna for a small size. However, two input connections are required to feed the mm-wave and microwave elements individually. In ref. [18], a combination of the SIW-based cross-slot and the annular ring antenna was proposed for dual-band application using the aperture coupling feeding mechanism. This design, however, necessitates a multilayer substrate, and the bandwidth at the upper band is only 1.3%. A shared aperture dual-band reflectarray and a patch antenna array for the S- and Ka-bands were designed in [19]. Dual-band operation is achieved by incorporating a low-frequency patch antenna array and a high-frequency reflectarray into the same aperture region. However, this is a multilayer structure; the impedance bandwidth is 200 MHz (6%) and 5.1 GHz (20%), and the peak realized gain is 13.70 dBi and 27.65 dBi at 3.5 GHz and 25.8 GHz, respectively.

In this paper, a low-profile, dual-band antenna with an integrated inset-fed upper band patch and a coupled-fed lower band patch antenna is presented, as depicted in Figure 1. The paper describes a novel design for a low-profile dual-band antenna that utilizes an inset-fed upper band patch and a coupled-fed lower band patch antenna, arranged in a planar configuration to make the design more compact. Therefore, the novelty of the antenna lies in its simplicity. While other works for V2V communication have come up with complicated designs to achieve similar results, we were able to work on optimizing the antenna while minimizing the complexity of the design, and we were still able to achieve favorable outcomes [11]. The modified U slot in this design ensures uniform gain for both patches, resulting in a >6 dB gain in both bands. The bandwidth of the upper band patch is doubled to 16% by creating parallel slots near the feed line. The achieved bandwidth is around 250 MHz for the lower band and 4.5 GHz for the 5G mmWave band. The implemented dual-band antenna offers high gain, frequency flexibility, and higher harmonic rejection between the two bands. Table 1 displays the specifications of the implemented dual-band antenna with enhanced bandwidth.

The paper is organized as follows: Section 2 explains the configuration and design procedure of the dual-band antenna and the modified dual-band antenna with enhanced bandwidth. In Section 3, the dual-band antenna with improved bandwidth was fabricated, measured, and compared with other dual-band antennas. Finally, Section 4 concludes the paper.

## 2. Design Procedures for the Dual-Band Antenna

### 2.1. Design I: Dual-Band Patch Antenna

The dual-band antenna is designed for the Rogers RT/Duroid 5800LZ with thickness t = 1.27 mm, dielectric constant $\epsilon_{r}$ = 2, and dielectric loss tangent tanδ = 0.0021. The total dimensions of the designed single-fed dual-band antenna are 22 mm × 26 mm × 1.27 mm. The antenna configuration consists of an upper band patch antenna embedded on the non-radiating edges of the lower band patch antenna, as displayed in Figure 1. First, a 28 GHz patch antenna with an inset feed line operating in TM10 mode is designed. The length and width of the 28 GHz patch are estimated using [20]. An inset feed line is introduced to obtain 50 Ω impedance matching. Next, a 5.9 GHz rectangular patch is designed with an optimized length $L_{pl}$ and width $W_{pl}$. Furthermore, the designed 28 GHz patch is added to the non-radiating edges of the 5.9 GHz patch with a gap for coupled feeding. As such, the lower band patch is excited by the coupling from the upper band patch.

The dual-band antenna was designed and simulated using a full-wave simulator. The surface current distribution of the designed dual-band antenna at the operating frequencies is first analyzed for a better understanding. First, the current distribution at 28 GHz (phase = 0) is examined, as shown in Figure 2a. The smaller patch controls the resonance in the upper band. Indeed, the upper band patch operates at its fundamental TM10 mode, while the lower band patch remains undisturbed. Similarly, Figure 2b illustrates the current distribution of the dual-band patch antenna at 5.9 GHz (phase = 90). It is clear that the current mainly flows over the lower band patch at 5.9 GHz through the coupling from the upper band patch. A time delay is indicated due to the direct excitation of the high-frequency antenna. When the antenna operates at 28 GHz, the coupled signal travels a specific distance, introducing a delay. Since the exact delay of the signal is unknown, a reference point, such as a peak or null point, or a phase of 90 degrees, can be considered. This reference point helps to account for the signal delay and appropriately align the phases.

The simulated S11 (dB) of the dual-band antenna (see Figure 3a) shows a −10 dB impedance bandwidth from 5.81 GHz to 5.99 GHz and 27.9 GHz to 30.1 GHz. Figure 3b shows the fundamental mode, TM10, and the higher-order mode, TM20, at 11.31 GHz. The upper patch in the design was specifically optimized for a frequency of 5.9 GHz, including its width and length parameters. As a result of the proper coupling and optimization of the 5.9 GHz patch, the higher-order harmonics at frequencies above 5.9 GHz are reduced. The space between the two patches was carefully optimized to suppress these harmonics. Notably, the optimized design of the upper patch helps mitigate the effects of harmonics and higher-order modes that could occur at other frequencies.

A parametric study was carried out to further analyze the patch lengths’ effect on the operating frequencies. Figure 4 demonstrates that adjusting the length of the lower band patch $L_{pl}$ shifts the lower band resonance. Similarly, the length of the upper band patch, $L_{pu}$, controls the upper band resonance, as shown in Figure 5. Therefore, the operating frequencies of the upper and lower bands can be easily tuned by controlling $L_{pu}$ and $L_{pl}$.

### 2.2. Design II: Dual-Band Patch Antenna

The bandwidth of the upper band patch can be further improved by adding parallel slits near the inset feed line, as shown in Figure 6. For the small patch, slits introduced additional resonance. It is also noted that the resonance frequency of this slot should be close to the resonance frequency of the patch. Therefore, the additional resonances were well coupled to the original patch resonance, enhancing antenna bandwidth in the smaller patch. Indeed, a bandwidth improvement from 8% to 16% is achieved by using this technique.

Antenna II operates from 5.8 GHz to 5.95 GHz and 26.8 GHz to 31.3 GHz, as shown in Figure 7. The simulated antenna provides a peak gain of 7.3 dBi in the DSRC band and 6.4 dBi in the mm-wave band. The normalized radiation patterns of antenna II at 5.9 GHz and 28 GHz in the E and H planes are shown in Figure 8.

As shown in Figure 2, the 28 GHz patch excites the 5.9 GHz patch. However, as seen in Figure 8, the 5.9 GHz patch radiates in the boresight direction, which means its radiation is not tilted from the broadside. The radiation pattern was not changed for 5.9 GHz because the optimized width and length parameters of the 5.9 GHz patch antenna ensure that it behaves similar to a single-ended patch antenna at lower frequencies. This means that at 5.9 GHz, the patch antenna can radiate efficiently with the desired radiation pattern. Conversely, the 28 GHz patch is dominant at a higher frequency. As shown in Figure 8, the 28 GHz also radiates in the broadside direction because the 5.9 GHz acts as a parasitic element for 28 the GHz. It is noted that the ground plane for 28 GHz is larger than the ground plane size for 5.9 GHz. That’s why the front-to-back ratio is high at 5.9 GHz, as shown in Figure 8a. The spacing between the two radiating patches is a crucial factor that significantly impacts the overall performance. The gap effect is shown in Figure 9 The radiation efficiency is displayed in Figure 10. The radiation efficiency ranges from 90.8% at 5.9 GHz to 98% at 28 GHz. As mentioned, the DSRC band ranges from 5.85 to 5.925 GHz for V2V communication. Similarly, our measured and simulated bandwidth, with 90.8% efficiency, ranges from 5.8 to 6.05 GHz, which falls within the DSRC band. For the small patch, the minimum radiation efficiency is around 94% and ranges from 26.8 to 31.8 GHz. An antenna can achieve high efficiency when its gain and directivity are closely aligned. In this case, the antenna demonstrates higher directivity at 28 GHz, as observed from the polar plot in Figure 8. This indicates that the antenna’s radiation pattern is more focused and concentrated in a specific direction at that frequency. Similarly, the antenna’s gain is also higher at 28 GHz, further indicating its effectiveness in capturing and transmitting signals in a more focused manner. The close correlation between gain and directivity at this frequency contributes to the overall efficiency of the antenna system.

## 3. Fabricated and Measured Dual-Band Antenna with Enhanced Bandwidth

### 3.1. Measured Results

The designed dual-band antenna with enhanced bandwidth (Design II) was fabricated and measured. The fabricated antenna prototype is shown in Figure 11a, and its measurement setup is shown in Figure 11b. The fabricated antenna was fed using a 2.4 mm end launch connector. The input reflection coefficient of the dual-band antenna was measured using a vector network analyzer, and its radiation patterns and gains were measured using an anechoic chamber.

Figure 12 illustrates the measured S11 (dB) of the 50 Ω impedance-matched dual-band antenna with increased bandwidth. The measured S11 agrees well with the simulated one, as displayed in Table 2. Moreover, the fabricated antenna provides a −10 dB impedance bandwidth from 5.8 GHz to 6.05 GHz, covering the DSRC band, and 26.8 GHz to 31.3 GHz, covering the 5G mm-wave band. Overall, the dual-band antenna provides a bandwidth of 250 MHz in the lower frequency band and a wide bandwidth of 4.5 GHz in the mm-wave band.

The measured radiation pattern of the dual-band antenna with enhanced bandwidth at 5.9 GHz is presented in Figure 13. The simulated and measured radiation patterns at 5.9 GHz agree well in both the E plane (ϕ = 0) and the H plane (ϕ = 90). Similarly, the fabricated dual-band antenna provides low cross-polarization levels at 5.9 GHz in both the E and H planes. The antenna offers a measured HPBW of 92° in the E plane and 102° in the H plane. A measured peak high gain of 7.7 dBi is observed in the lower frequency band, as shown in Figure 14. The antenna has a directional radiation pattern in both frequency bands.

The measured radiation pattern of the fabricated antenna at 28 GHz is displayed in Figure 15. The discrepancies between the measured and simulated patterns were due to fabrication errors and the presence of the connector for the mm-wave frequency. The feedline for the 28 GHz patch is very thin; hence, the soldering needed careful attention close to the feedline. Moreover, the fabricated antenna was fed using a 2.4 mm end launch connector to measure the radiation. As a result of the connector’s cable loss, there are few discrepancies between the measured and simulated patterns. Figure 15 also shows that the dual-band antenna at 28 GHz provides an HPBW of 22° in the E plane and 63° in the H plane. A measured peak gain of 6.38 dBi is observed in the upper-frequency band, as shown in Figure 14. At 28 GHz, the cross-polarization levels in both the E and H planes are acceptable. The cross-polarization in the upper band can be further reduced by adding defective ground structures (DGS). Implementing DGS helps optimize the antenna’s performance by suppressing unwanted polarization and enhancing the desired polarization. This integration improves overall system performance and reduces cross-polarization effects during communication.

### 3.2. Comparison with Other Dual-Band Antennas

This section compares the implemented dual-band antenna with other dual-band antennas with a large frequency ratio, as shown in Table 3. The comparison focuses on several key aspects: frequency tunability, harmonic reduction, bandwidth improvement, and gain. While the exact frequency is not the primary concern, the goal is to achieve an antenna with a high frequency ratio with the mentioned features. The gain of the designed antenna at both bands is higher than in other published works. The antenna in [12] rejects higher-order harmonics between the two bands. However, it requires an additional compact microstrip resonant cell (CMRC) low-pass filter design. The antennas in refs. [11,21] were compact, but they suffer from low gain at the lower bands and require the additional design of a CMRC low-pass filter. The antenna in ref. [11] provides good gain at both frequencies but requires a complicated multilayer design and expensive fabrication techniques. In contrast, the implemented dual-band antenna with enhanced bandwidth is simple to design and does not require a complex feeding structure. Additionally, the designed antenna rejects higher-order harmonics between the bands, provides high gain, and is compact. The dimensions of the antennas in millimeters, along with a visual representation of the structures, are shown in Figure 16.

## 4. Conclusions

This paper presents a single-layer, single-feed, dual-band antenna with enhanced bandwidth. The designed antenna consists of an inset-fed 28 GHz patch embedded into the 5.9 GHz patch to provide dual-band operation. Furthermore, the bandwidth at the upper-frequency band is improved from 8% to 16% by adding parallel slots near the inset-feed line. The fabricated antenna operates from 5.8 GHz to 6.05 GHz and 26.8 GHz to 31.3 GHz, covering the DSRC and 5G mm-wave bands. Overall, the designed dual-band antenna offers the following:Frequency tunability: This antenna design proposes a frequency-tunable dual-band patch antenna with a high-frequency ratio that can be tuned between 5.9 GHz and 28 GHz.Reduced harmonics between the two bands: The design was optimized for a frequency that included the width and length parameters for the lower band patch. By optimizing the upper patch (lower band patch) design for 5.9 GHz, it was found that the higher-order harmonics at frequencies above 5.9 GHz were reduced because of the lack of proper coupling. Therefore, optimizing the design of the upper patch is a tool that can be used to reduce the effects of harmonics and higher-order modes that could occur at other frequencies.Bandwidth improvement at the upper band: slits introduced additional resonance for the small patch. As a result, the additional resonances were well-coupled to the patch resonance, enhancing antenna bandwidth. In our design, the bandwidth at the upper-frequency band is improved from 8% to 16% by adding parallel slots near the inset-feed line.High gain at both bands: The designed dual-band antenna offers a high peak gain of 7.7 dBi in the DSRC band and 6.38 dBi in the mm-wave band. Both antennas radiate in the boresight direction.; andSingle-layer structure, simple design, and low-cost PCB fabrication.

The FCC is modernizing the 5.9 GHz frequency to improve Wi-Fi and automotive safety. The 28 GHz band, within the Ka-band, is utilized by the satellite industry. Both frequencies are also relevant to implementing 5G networks, covering the FR1 and FR2 bands. These frequencies play a crucial role in advancing wireless communication, satellite services, and the development of high-speed connectivity.

## Figures and Tables

**Figure 1 sensors-23-06108-f001:**
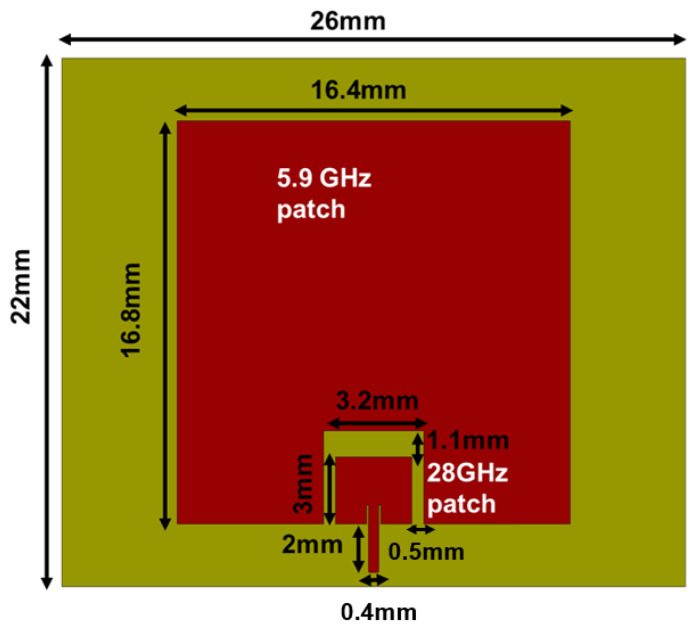
Design I: Dual-band patch antenna operating at 5.9 GHz and 28 GHz.

**Figure 2 sensors-23-06108-f002:**
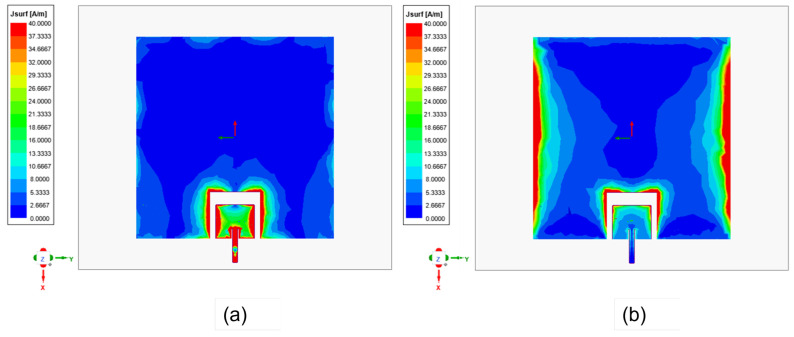
Surface current distribution at (**a**) 28 GHz, phase = 0 and (**b**) 5.9 GHz, phase = 90.

**Figure 3 sensors-23-06108-f003:**
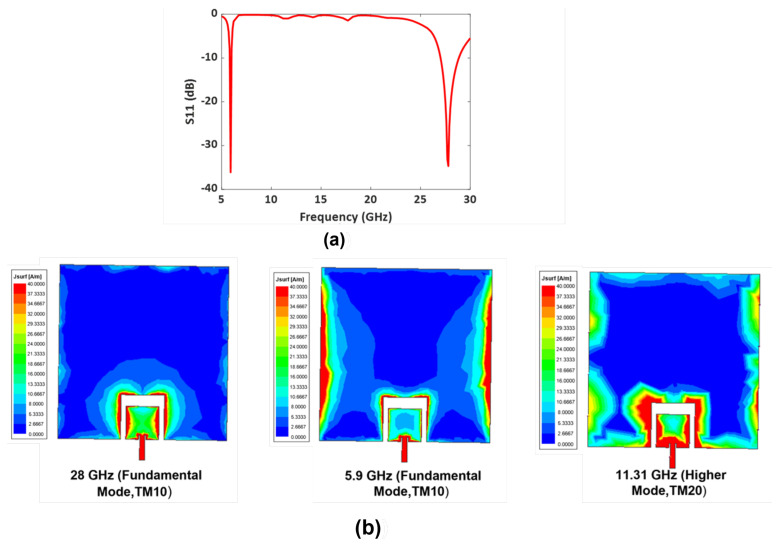
(**a**) Simulated S11 (dB) of Design I: Dual-band antenna operating at 5.9 GHz and 28 GHz. (**b**) Antenna with fundamental mode and higher modes.

**Figure 4 sensors-23-06108-f004:**
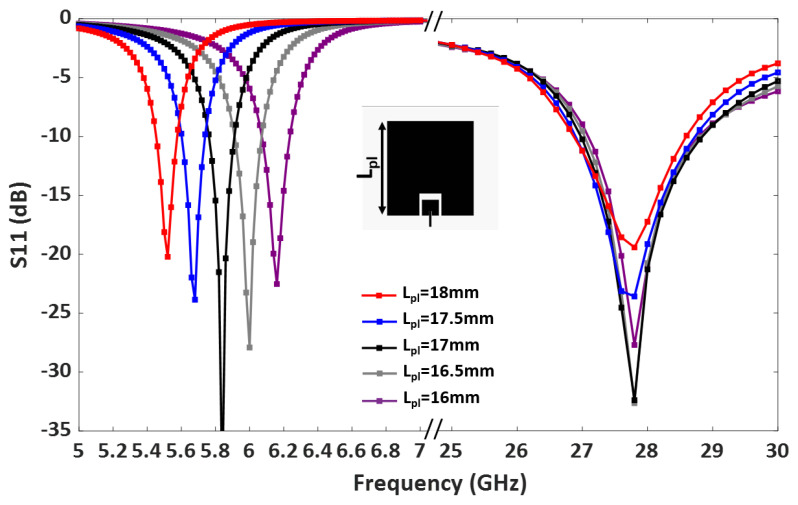
Parametric study of the dual-band antenna by adjusting the length of the lower band patch, $L_{pl}$.

**Figure 5 sensors-23-06108-f005:**
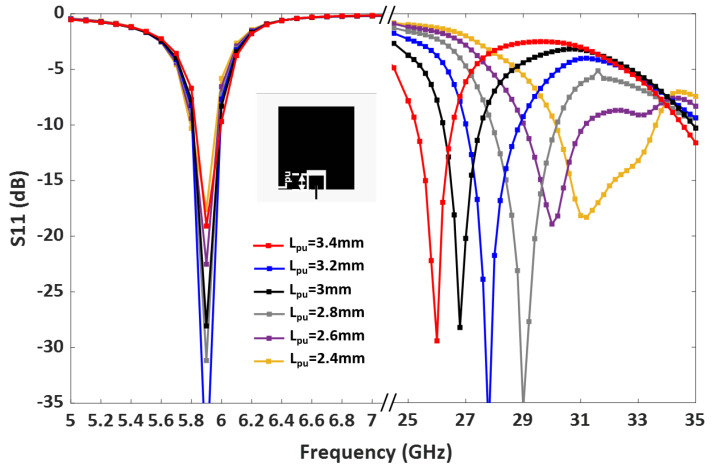
Parametric study of the dual-band antenna by adjusting the length of the upper band patch, $L_{pu}$.

**Figure 6 sensors-23-06108-f006:**
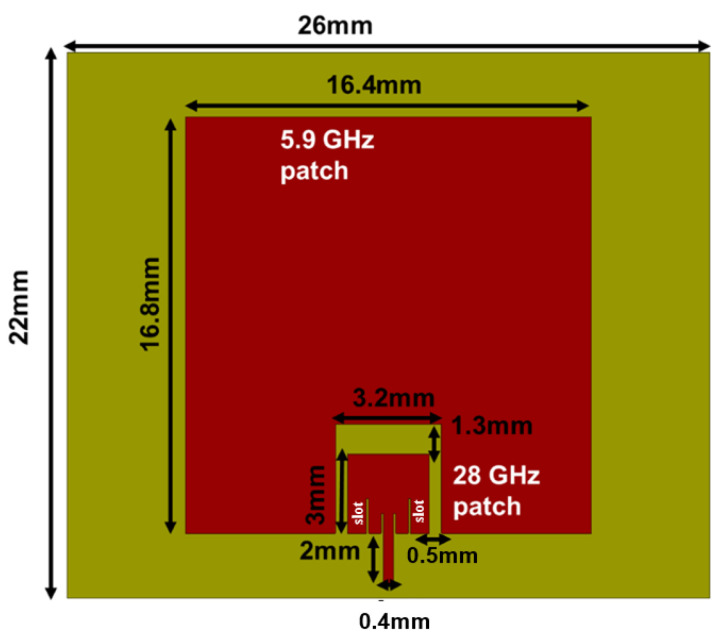
Configuration of the Design II: Dual-band antenna with enhanced bandwidth with parallel slots near the inset feed line.

**Figure 7 sensors-23-06108-f007:**
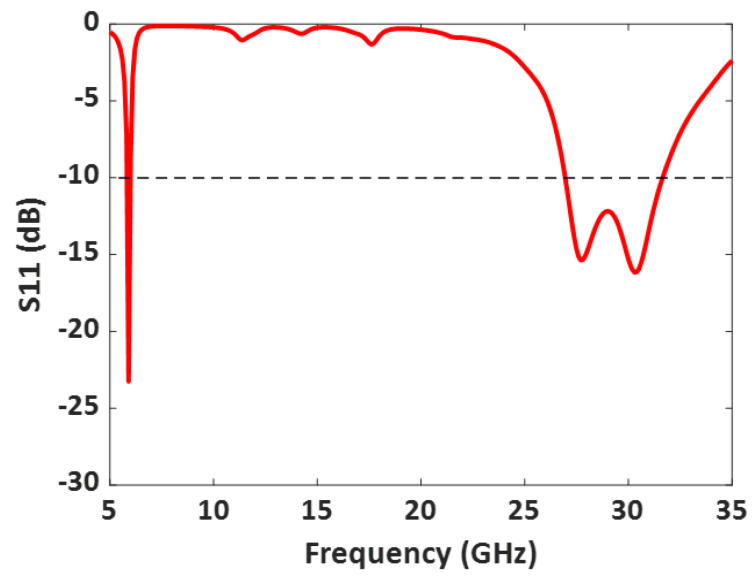
Simulated S11(dB) of the Design II: Dual-band antenna with enhanced bandwidth with parallel slots near the inset feed line.

**Figure 8 sensors-23-06108-f008:**
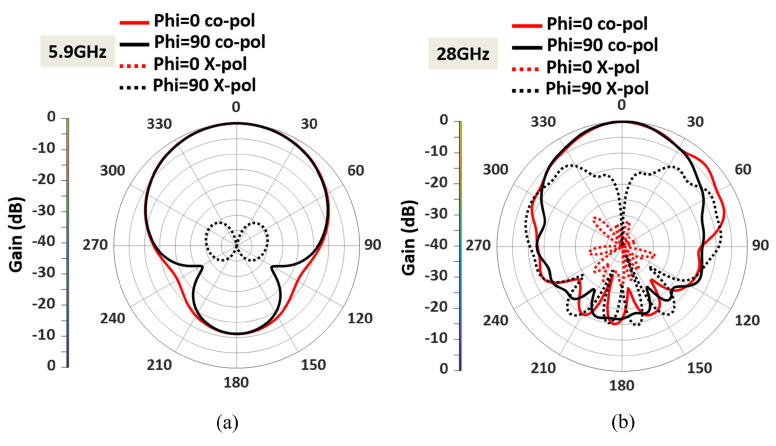
Simulated radiation pattern of the dual-band antenna with enhanced bandwidth within both E (XZ) and H (YZ) planes at (**a**) 5.9 GHz, and (**b**) 28 GHz.

**Figure 9 sensors-23-06108-f009:**
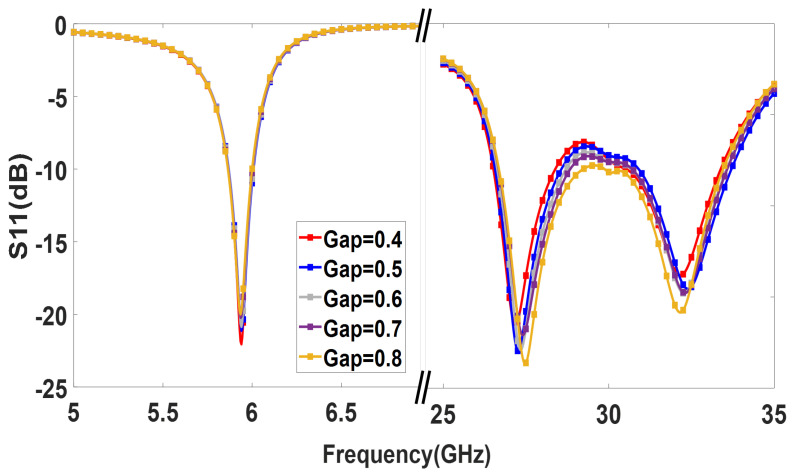
Gap effect among the radiating patch.

**Figure 10 sensors-23-06108-f010:**
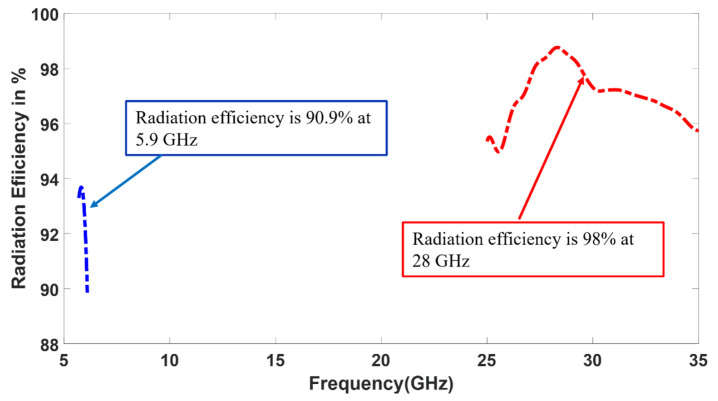
Simulated Radiation efficiency of the dual-band antenna with enhanced bandwidth at 5.9 GHz and 28 GHz.

**Figure 11 sensors-23-06108-f011:**
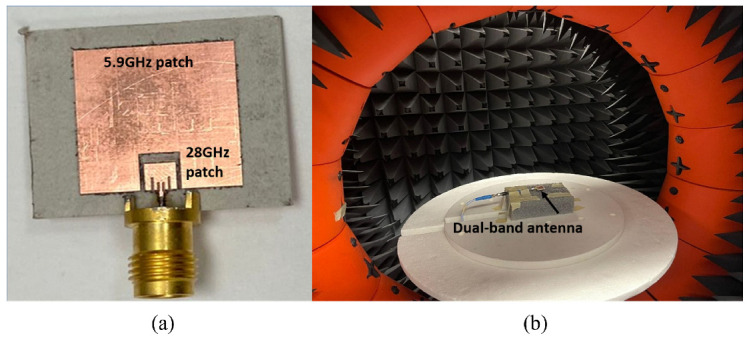
(**a**) Fabricated dual-band antenna (**b**) its measurement setup.

**Figure 12 sensors-23-06108-f012:**
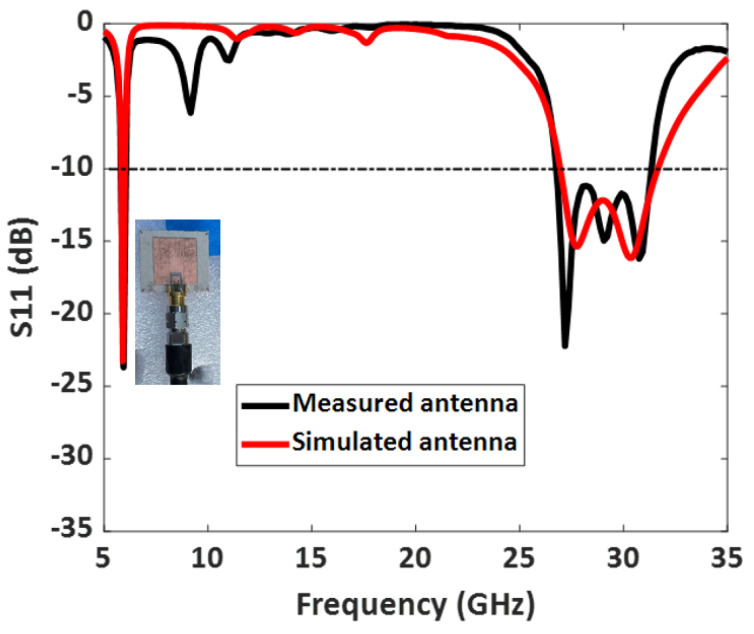
Measured and simulated S11 (dB) of the fabricated dual-band antenna with enhanced bandwidth.

**Figure 13 sensors-23-06108-f013:**
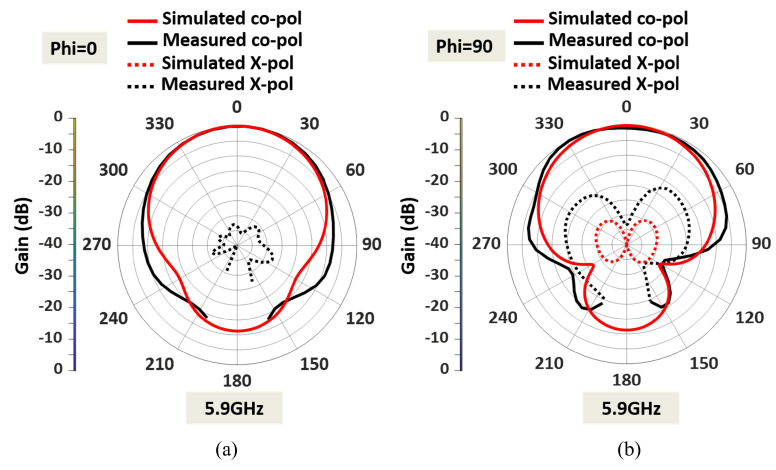
Normalized measured and simulated radiation patterns at lower frequency band 5.9 GHz in the (**a**) E plane (XZ plane), and (**b**) H plane (YZ plane).

**Figure 14 sensors-23-06108-f014:**
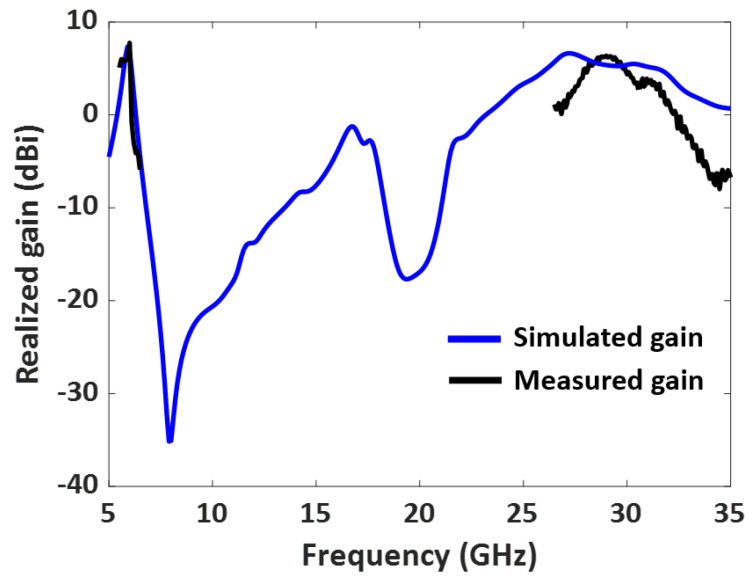
Measured and simulated gain (dBi) vs. frequency of the designed dual-band antenna.

**Figure 15 sensors-23-06108-f015:**
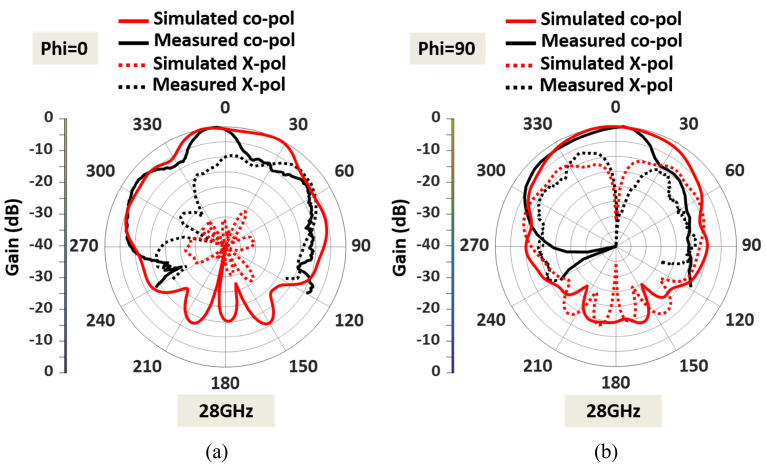
Normalized measured and simulated radiation patterns at upper band 28 GHz (**a**) E plane (XZ plane), and (**b**) H plane (YZ plane).

**Figure 16 sensors-23-06108-f016:**
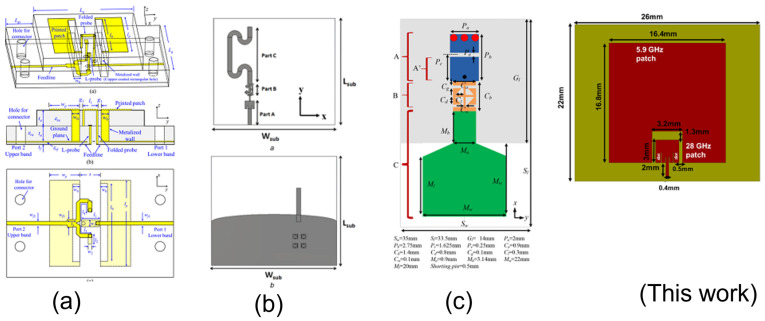
Visual representations of the antenna structures (**a**) [11], (**b**) [12], (**c**) [21], and This work presented in Table 3.

**Table 1 sensors-23-06108-t001:** Specifications of the dual-band antenna.

	DSRC Band	5G mm-Wave Band
**Frequency**	5.9 GHz	28 GHz
**Impedance Bandwidth**	250 MHz	4.5 GHz
**Polarization**	Linear polarization	
**Peak Gain**	7.7 dBi	6.38 dBi
**HPBW**	92°/102°	22°/63°
HPBW = Half Power Beamwidth		

**Table 2 sensors-23-06108-t002:** Measured and simulated S11 <−10 dB of Design II: Dual-band antenna with enhanced bandwidth.

	DSRC Band	5G mm-Wave Band
**Simulated Frequency (GHz)**	5.81–5.99 GHz	26.9–31.3 GHz
**Measured Frequency (GHz)**	5.8–6 GHz	26.8–31.6 GHz

**Table 3 sensors-23-06108-t003:** Comparison with other state-of-the-art dual-band antennas with large frequency ratio.

Reference	[11]	[12]	[15]	[21]	This Work
**Center Frequency** **(GHz)**	5.9 /28	2.4 /5.5 /28	2.4 /28	2.45 /60	**5.9 /28**
**Frequency Ratio**	4.74	11.6	11.6	24.4	**4.74**
**Operating Bandwidth**	47.1% /6.67%	15.4% /22.2% /11.4%	2.6:1 /8.9%	2.5:1 /11.6%	**4.2% /16%**
**Dimensions ($\lambda_{l}$ × $\lambda_{l}$ × $\lambda_{l}$ ** ${}^{a}$ **)**	0.65 × 0.43 × 0.12	0.36 ×0.32 × 0.004	-	0.28 ×0.38 × 0.002	**0.43 × 0.51 × 0.025**
**Peak Gain (dBi)**	5.8/8.46	1.95 /3.76 /7.35 ${}^{b}$	2.6 /6.5	3 /6	**7.7 /6.38**
**No. of Ports**	2	1	1	1	**1**
**Harmonics Rejection**	No	Yes	No	No	**Yes**

${}^{a}$ Wavelength of the center frequency of the lower frequency band. ${}^{b}$ Simulated results.

## Data Availability

Data is protected due to privacy or ethical restrictions.

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
