# Peer review of "Modified U Slot Patch Antenna with Large Frequency Ratio for Vehicle-to-Vehicle Communication"

_sensors, 2023, doi:10.3390/s23136108_

Round 1

Reviewer 1 Report

In this paper, authors presents a single-fed, single-layer, dual-band antenna for Vehicle-to-Vehicle communication.

The antenna is very conventional and these kinds of antennas have been widely studied in the literature. The  novelty and contribution of this work is very limited.

However, this is an article about antenna design, for this type of article it is written correctly and can be published after introducing a few changes.

Several elements need to be corrected in the paper:

(1)      The paper is well written and the presentation of the work is good.  However, the Figures quality is poor which needs improvement in the revised version.

(2)      Figure 1 and Figure 6 of poor quality, please change the colors.

(3)      The authors use the term "dual-wideband antenna". With a bandwidth of 16%, we are not talking about a wideband antenna, please correct the terminology.

(4)      In Figure 3, please present the results of the S11 reflection coefficient simulation so that they are legible.

(5)      The authors specify the bandwidth in the paper in many places (abstract, tables, text) and there is no consistency here. Different values are given, please correct.

(6)      A lot of editing errors, e.g. S11, when numbering chapters or bullet points in the summary. Please correct

(7)      The performance comparison presented in this manuscript is not fair. The authors must compare their work with other antennas operating at the same frequencies.

(8)      Table 3: the comparison would be more reliable if the table included the dimensions of the antennas in mm and, for example, a drawing of the shape - I suggest introducing such a change (the article will benefit from it).

(9)      Radiation patterns: there is no definition of the reference system (coordinate system) in which the polarization of the antenna is determined.

(10)   The authors in the conclusions may add an exemplary practical application of the proposed antenna.

Please have the article read by a native speaker so that he or she can correct any language errors.

Author Response

 PDF is attched.

Reviewer 2 Report

This paper presents dual-band antenna. There are several comments to improve the paper as follows:

1. The harmonic suppression mechanism is not mentioned. The authors just claimed that the higher-order harmonics are reduced due to the lack of proper coupling, which definitely is not the explanation. Pls discuss about this.

2. In Fig. 2, why the phase at 5.9 is 90 deg?

3. In Fig. 3b, what are the phases of antenna operating at 28 GHz and 5.9 GHz?

4. The most important parameters are the gaps between the two radiating patches. Pls show the effect of these parameters.

5. The front-to-back ratio is also imprtant. According to Fig . 8a, the back radiation at 5.9 GHz is quite high. Pls give the reason and hows to reduce the back radiation.

6. The antenna efficiencies are very high, even at 28 GHz. Normarly, the patch antenna has efficiency of 85-90%. Pls explain this.

7. In measurement, whats type of SMA connector?

8. In Fig. 14, the cross polaization at 28 GHz is very high. It will cause strong negative effect on the communication system. The proposed antenna is designed for practical applications. Thus, the technical requirements should be satisfied.

The english is good.

Author Response

 PDF is attached.

Round 2

Reviewer 1 Report

All comments have been corrected. In my opinion, the article can be published in this form.

Reviewer 2 Report

The authors have well addressed all concerns.

Good